# The Improvement of Adaptive Immune Responses towards COVID-19 Following Diphtheria–Tetanus–Pertussis and SARS-CoV-2 Vaccinations in Indonesian Children: Exploring the Roles of Heterologous Immunity

**DOI:** 10.3390/vaccines12091032

**Published:** 2024-09-09

**Authors:** Theresia Santi, Juandy Jo, Alida Roswita Harahap, Retno Asti Werdhani, Sri Rezeki S. Hadinegoro, Ivo Novita SahBandar, Ari Prayitno, Zakiudin Munasir, Yvan Vandenplas, Badriul Hegar

**Affiliations:** 1Doctoral Program in Medical Science, Faculty of Medicine, Universitas Indonesia, Jakarta 10430, Indonesia; alida.roswita@ui.ac.id; 2Department of Biology, Faculty of Health Sciences, Universitas Pelita Harapan, Tangerang 15811, Indonesia; juandy.jo@uph.edu; 3Mochtar Riady Institute for Nanotechnology, Tangerang 15811, Indonesia; 4Department of Community Medicine, Cipto Mangunkkusumo Hospital, Faculty of Medicine, Universitas Indonesia, Jakarta 10430, Indonesia; retno.asti@ui.ac.id; 5Department of Child Health, Cipto Mangunkusumo Hospital, Faculty of Medicine, Universitas Indonesia, Jakarta 10430, Indonesia; shadinegoro46@gmail.com (S.R.S.H.); ariprayitno@ikafkui.net (A.P.); zakiudin@ikafkui.net (Z.M.); 6Department of Microbiology, School of Medicine, Iwate Medical University, Morioka 028-3694, Japan; ivosb@hawaii.edu; 7Department of Pediatric, Universitair Ziekenhuis Brussel, 1090 Jette, Belgium; yvan.vandenplas@uzbrussel.be; 8Indonesian Medical Education and Research Institute, Faculty of Medicine, Universitas Indonesia, Jakarta 10430, Indonesia

**Keywords:** children, SARS-CoV-2-specific immune response, heterologous immunity, diphtheria–tetanus–pertussis vaccine, inactivated COVID-19 vaccine

## Abstract

Background: Routine childhood vaccination, e.g., for diphtheria, tetanus, and pertussis (DTP), might provide additional protection against SARS-CoV-2 infection. This concept of heterologous immunity was explored in healthy children receiving both DTP and inactivated SARS-CoV-2 vaccines. Methods: A cross-sectional study was performed on 154 healthy children aged 6–8 years old in Jakarta, Indonesia. Their vaccination status for the DTP (including a diphtheria–tetanus booster vaccine at 5 years old) and CoronaVac (from 6 years old) vaccines were recorded. Peripheral blood samples were collected from all participants, in which anti-diphtheria toxoid IgG and anti-SARS-CoV-2 S-RBD antibodies and T cell-derived IFN-γ were measured. Results: The study participants with complete DTP vaccination had significantly higher titers of anti-diphtheria toxoid IgG than the ones without (median = 0.9349 versus 0.2113 IU/mL; *p* < 0.0001). Upon stratification based on DTP and CoronaVac vaccination statuses, the participants with complete DTP and CoronaVac vaccinations had the highest titer of anti-SARS-CoV-2 S-RBD antibodies (median = 1196 U/mL) and the highest concentration of SARS-CoV-2-specific T cell-derived IFN-γ (median = 560.9 mIU/mL) among all the groups. Conclusions: Healthy children aged 6–8 years old with complete DTP and CoronaVac vaccinations exhibited stronger SARS-CoV-2-specific T cell immune responses. This might suggest an additional benefit of routine childhood vaccination in generating protection against novel pathogens, presumably via heterologous immunity.

## 1. Introduction

The coronavirus disease 2019 (COVID-19), caused by severe acute respiratory syndrome coronavirus 2 (SARS-CoV-2) infection, is now globally endemic. From early 2020 until June 2024, the cumulative number of COVID-19 cases in the world and Indonesia has risen to 775,583,309 and 6,829,057 confirmed cases, respectively [1]. Furthermore, the COVID-19-associated mortality in the world and Indonesia by early June 2024 was already at 7,050,691 and 162,058 deaths, respectively [1], emphasizing the need for effective prevention and treatment methods for COVID-19. With regard to preventive measures, vaccination is one of the most important public health tools. Various types of vaccines are available against symptomatic SARS-CoV-2 infection, including, but not limited to, inactivated virus vaccines (Sinovac’s CoronaVac, Sinopharm’s Covilo BBIBP-CorV, and Bharat Biotech’s Covaxin), messenger RNA-based vaccines (Moderna’s Spikevax mRNA-1273 and Pfizer–BioNTech’s Comirnaty BNT162b2), adenovirus vector–based vaccines (AstraZeneca’s Vaxzevria and Covishield ChAdOx1 as well as Johnson & Johnson–Janssen’s Ad26.COV2.S), and adjuvanted protein vaccines (Novavax’s Nuvaxovid and Covovax NVX-CoV2373) [2]. The widespread disparity in vaccine distribution, however, impedes the collective effort to mitigate the severity of COVID-19 by administering the vaccines, with populations from low- and middle-income countries disproportionately suffering due to this inequality [3,4].

One sub-group of those populations is children under 12 years old. This age group is less prioritized in many countries for vaccination because of the limited availability of COVID-19 vaccines, the presumed milder symptoms of SARS-CoV-2 infection among children, as well as the incomplete knowledge on the risks and benefits offered by pediatric COVID-19 vaccines [5,6,7,8,9]. However, this does not exclude a risk that SARS-CoV-2 infection could cause a severe disease or multisystem inflammatory syndrome in children (MIS-C) [10,11,12]. Indeed, several low- and middle-income countries, including Indonesia, have reported relatively high incidences of severe COVID-19 among children, particularly prior to the circulation of the Omicron variant [13,14,15,16,17], stressing an urgent need for COVID-19 vaccination for this age group in these countries. In addition, SARS-CoV-2-infected children, despite their asymptomatic status, could become a potential reservoir for transmission to vulnerable family members and the evolution of SARS-CoV-2 variants [18]. Taken together, policies to vaccinate children under 12 years old against SARS-CoV-2 infection are arguably prudent.

Various types of COVID-19 vaccines have been used to vaccinate healthy children under 12 years old, including the mRNA-based (e.g., BNT162b2 and mRNA-1273), recombinant adenovirus vector-based (e.g., ChAdOX1), and inactivated SARS-CoV-2 vaccines (e.g., CoronaVac and BBIBP-CorV) [19,20]. To date, most studies on COVID-19 vaccination in children focused on the administration of the BNT162b2 and CoronaVac vaccines as these two vaccines are arguably the most common COVID-19 vaccines used in children [20,21]. Nonetheless, a recent meta-analysis was performed to assess the immunogenicity and effectiveness of all the COVID-19 vaccines (including BNT162b2 and CoronaVac) used in children and adolescents; it reported that those who were vaccinated with the mRNA-based vaccines had a higher seroconversion rate than the those who received inactivated vaccines (98.8 versus 92.8%). Interestingly, the seroconversion rate was higher in vaccinated children aged 5–11 years than in adolescents aged 12–18 years (97.6 versus 91.3%) [20]. This study also calculated the pooled vaccine effectiveness (by combining the results of all the vaccines) and found that the fully vaccinated group exhibited higher effectiveness against SARS-CoV-2 infection (63.3 versus 42.9%), COVID-19 (75.8 versus 60.7%), and hospitalization due to COVID-19 (82.8 versus 72.7%) compared to the partially vaccinated group. Of note, within the fully vaccinated group, the pooled vaccine effectiveness of the COVID-19 vaccines against hospitalization due to COVID-19 (82.8%) was higher than the effectiveness against SARS-CoV-2 infection (63.3%) or against COVID-19 (75.8%). Intriguingly, the pooled vaccine effectiveness against SARS-CoV-2 infection (66.8 versus 38.7%), COVID-19 (74.9 versus 59.6%), as well hospitalization due to COVID-19 (90.1 vs. 65.9%) were higher in children and adolescents aged 12–18 than in children under 11 years, suggesting that the efficacies of COVID-19 vaccines in children aged 5–11 years old might not be as high as the ones observed in adolescents [20].

Upon administration of a new vaccine to children, parental concerns are commonly observed regarding safety and potential adverse events. For the mRNA-based vaccines, rare instances of myocarditis and pericarditis have been observed in adolescents and young adults, which appear to be dependent on the dosage and timing. Nonetheless, the risk of myocarditis in children aged six months to five years is expected to be minimal, considering the lower vaccine doses for the younger age group and the relatively low rates of myocarditis in young children. Other adverse events have been recorded as being lower in children under 11 years than in adolescents aged 12 to 17 years, with mild local and systemic reactions being the most reported ones, including injection site pain, tiredness, headache, myalgia, and chills [22,23,24,25]. For the inactivated SARS-CoV-2 vaccines, the adverse events were generally less pronounced in younger children, consistent with the observations for the mRNA-based vaccines [19,24,26,27], suggesting that the safety and tolerability profiles of the inactivated vaccines are acceptable. Of note, a subgroup analysis showed that the incidence rate of adverse events upon administration of the mRNA vaccines was higher than that of the inactivated SARS-CoV-2 vaccines [20]. In addition, over 65% of the side effects from COVID-19 vaccination in children resolved or diminished within 1–3 days [19]. Taken together, these reinforce the notion that the current COVID-19 vaccines, particularly the mRNA-based and inactivated vaccines, are safe to be prescribed for healthy children.

As the clinical trials on COVID-19 vaccines mainly tested healthy pediatric populations, there is an interest to assess the immunogenicity and safety of COVID-19 vaccines in pediatric patients with underlying chronic diseases, including immunodeficiencies. Although it was observed that immunocompromised children did not have a higher risk of contracting COVID-19 compared to healthy controls, high-risk immunocompromised pediatric patients tended to contract severe COVID-19, including the ones receiving intense chemotherapy or high doses of steroid medications [28]. As expected, children receiving immunosuppressive agents exhibited lower immunogenicity than immunocompetent pediatric patients upon vaccination with two doses of BNT162b2 or CoronaVac, suggesting that they would require additional vaccinations to confer sufficient protection against SARS-CoV-2 infection [29,30,31,32,33,34]. Nonetheless, both the BNT162b2 and CoronaVac vaccines were observed to be safe for pediatric patients with underlying diseases [29,30,33].

The Ministry of Health of Republic of Indonesia has recommended two COVID-19 vaccines for children aged 6–11 years old: CoronaVac and BNT162b2. However, the pediatric COVID-19 vaccination program in Indonesia is facing significant hurdles, including parental hesitancy and a shortage of vaccines [35,36]. As a part of the national COVID-19 vaccination program, the CoronaVac vaccine has been administered to Indonesian children aged 6 years old and above since December 2021. Until 28 August 2024, there were approximately 26.4 million children aged 6–11 years old who had received two doses of CoronaVac [37]. However, its availability has become scarce in Indonesia since October 2022 [38,39], prompting the need for improvised methods to keep protecting the pediatric population in Indonesia against SARS-CoV-2 infection.

One of the proposed methods is by investigating the protection against COVID-19 offered by routine vaccines against unrelated pathogens. This concept is known as heterologous immunity, which is an induction of adaptive immune responses using a certain pathogen or vaccine against an unrelated pathogen [40]. Three vaccines have been hypothesized to generate heterologous immunity due to their association with reductions in all-cause infant mortality: bacillus Calmette–Guérin, measles-containing vaccines, as well as the diphtheria–tetanus–pertussis (DTP) vaccine. DTP vaccination is of interest due to several reasons: (i) an in silico investigation reported high similarities between the antigens targeted by the SARS-CoV-2 and DTP vaccines [41]; (ii) diphtheria and tetanus vaccination has been associated with a lower risk of COVID-19 hospitalization among older populations in the UK [42]; and (iii) DTP booster vaccination (using a diphtheria–tetanus vaccine in Indonesia) is administered at 5 years old [43], which is a relatively short time before the start of CoronaVac vaccination at 6 years old. If heterologous immunity does exist, the immunological memory created by DTP vaccination and boosters will increase the efficacy of CoronaVac vaccination.

We therefore investigated the potential heterologous immunity induced by DTP (including diphtheria-tetanus booster) and CoronaVac vaccinations in healthy children aged 6–8 years old living in Jakarta, Indonesia. Peripheral blood samples were collected from all the study participants. Anti-diphtheria immunoglobulin G was quantified as an indicator of successful DTP vaccination. SARS-CoV-2-specific adaptive immune responses were assessed as well, i.e., the levels of anti-SARS-CoV-2 S-RBD antibodies and T cell-derived interferon gamma were measured to reflect the humoral and cellular immune responses, respectively.

## 2. Materials and Methods

### 2.1. Study Design and Participants

A cross-sectional study on healthy children aged 6–8 years old was conducted between November 2022 and October 2023, which was preceded by questionnaire-based data collection from parents of eligible subjects living in the Senen district, Central Jakarta, Indonesia. The questionnaire-based results have been published [36]. Pertaining to the status of DTP vaccination among the study participants, their titers of anti-diphtheria toxoid antibodies have been published as well (manuscript was accepted).

The inclusion criteria were (i) healthy children aged 6–8 years old and (ii) no confirmed history of COVID-19. The exclusion criteria were (i) obese and having a poor nutrition status; (ii) received less than three doses of the primary DTP immunization; (iii) received less than two doses of a COVID-19 vaccine (CoronaVac vaccine, an inactivated SARS-CoV-2, or aluminum hydroxide-adjuvanted vaccine [Sinovac Life Sciences, Beijing, China]); (iv) received a different type of immunization less than a month prior to the commencement of the study; (v) have primary immunodeficiency disease, autoimmune disease, cancer, or a chronic or congenital disease; and (vi) took medication that could alter the immune response in the past 4 weeks, e.g., long-term corticosteroid, intravenous immunoglobulin drugs, or blood products. The minimum number of subjects was estimated to be 110 individuals. This study was approved by the Ethics Committee of the Faculty of Medicine, Universitas Indonesia and Cipto Mangunkusumo Hospital (KET-1160/UN2.F1/ETIK/PPM.00.02/2022). All parents of the recruited study participants provided their consent for their children to participate.

This study was conducted following the Ministry of Health’s mandatory national vaccination program for school-age children in Indonesia for DTP and COVID-19 vaccinations. The DTP vaccine (using whole-cell pertussis) as well as DT booster vaccine were produced by BioFarma, Indonesia, in which the concentrations of diphtheria, pertussis, and tetanus were 30, 4, and 40 IU, respectively, per dose. The CoronaVac vaccine was produced by Sinovac Life Sciences, China, in which its concentration was 3 μg per dose. In this routine vaccination program for school-age children, the vaccines were distributed via the Province Deputy of the Ministry of Health and were transported and stored strictly adhering to the recommended cold chain management. Trained medical staff intramuscularly administered the vaccines using a sterile 0.5 mL syringe to the deltoid area of the children. An aseptic procedure was always properly performed prior to the administration. Parents were provided with letters advising them on how to manage any adverse reactions post-immunization, such as fever or pain at the site of injection [44]. It is well known that upon DTP (using whole-cell pertussis) vaccination, mild side effects are commonly observed in infants and children, including local reactions (50%) and systemic reactions, e.g., fever above 38 °C and irritability (40–75%), drowsiness (33–62%), loss of appetite (20–35%), and vomiting (6–13%) [45]. Severe adverse events upon DTP vaccination (using whole-cell pertussis) are uncommon, but they could include a temperature exceeding 40.5 °C (occurring in 0.3% of vaccine recipients), febrile seizures (8 per 100,000 vaccinated subjects), or hypotonic–hyporesponsive episodes (ranging from 0 to 291 per 100,000 vaccinated subjects) [45]. No serious adverse events following CoronaVac vaccination among children have been reported in Indonesia to date. Of note, the timing of the administration of the DTP vaccine, DT booster vaccine, and CoronaVac vaccine were obtained from the official vaccination records.

### 2.2. Humoral Immune Response Assays

The blood collected from all the participants was assessed for serum anti-diphtheria immunoglobulin G (IgG) and SARS-CoV-2 S-RBD antibodies. The titers of anti-diphtheria IgG were assessed using a commercial enzyme-linked immunosorbent assay (ELISA) (anti-diphtheria toxoid IgG ELISA, EUROIMMUN, Lübeck, Germany). This method utilizes inactivated diphtheria toxin as a tracer antigen for the quantitative assessment of anti-diphtheria toxoid IgG in serum [46]. The total antibodies (including IgG) against the SARS-CoV-2 spike protein receptor-binding domain (S-RBD) were assessed using an electrochemiluminescence assay from Roche (Elecsys^®^ anti-SARS-CoV-2) with the Cobas e 411 analyzer (Roche Diagnostic, Rotkreuz, Switzerland). According to the manufacturer, the measuring range is between 0.4 and 250 U/mL in undiluted samples, with a value below 0.8 U/mL considered negative and a value equal to or above 0.8 U/mL considered positive [47]. If an undiluted sample exhibited a value above 2500 U/mL, it would be diluted 1:10 until it exhibited a value below 2500 U/mL.

### 2.3. Cellular Immune Response Assay

The specific cellular immunity was assessed by measuring SARS-CoV-2-specific T cell-derived interferon gamma (IFN-γ) levels using the EUROIMMUN SARS-CoV-2 Quan-T cell interferon gamma-release assay (IGRA) (EUROIMMUN, Lübeck, Germany). The assay was used to assess the specific production of IFN-γ by CD4^+^ and CD8^+^ T lymphocytes upon stimulation with SARS-CoV-2 spike protein-derived peptides [48]. Briefly, 500 μL of whole blood in a lithium heparin tube was transferred within an hour to three tubes: (i) a SARS-CoV-2 IGRA BLANK with no T cell activating component, reflecting the background T cell activity, (ii) a SARS-CoV-2 IGRA TUBE coated with a pool of SARS-CoV-2 spike protein-derived peptides that are able to stimulate specific CD4^+^ and CD8^+^ T lymphocytes, and (iii) a SARS-CoV-2 IGRA STIM tube coated with a mitogen for non-specific T cell stimulation, which was used as a control for the viability and stimulation ability of the T cells. After six inversions, these tubes were incubated at 37 °C for 20–24 h and subsequently centrifuged at room temperature for 10 min at 12,000× *g*. The supernatant was stored at −20 °C until the measurements were performed. The IFN-γ measurements were performed using ELISA (EUROIMMUN, Lübeck, Germany). The supernatant was diluted 1:5 with a dilution buffer. Six calibrators and two controls were used in each run. The specific IFN-γ concentrations were obtained after subtracting the BLANK value from the TUBE/STIM value, and the concentrations were expressed in mIU/mL. Results < 100 and >200 mIU/mL were considered negative and positive, respectively. The upper limit of detection was 2500 mIU/mL.

### 2.4. Statistical Analysis

The statistical analyses were performed using the IBM SPSS Statistics for Windows version 26.0 (IBM Corp., Armonk, NY, USA). Descriptive data are presented as median values with minimum and maximum values for continuous variables, while categorical variables are presented as a frequency and percentage. Differences in anti-diphtheria toxoid IgG or anti-SARS-CoV-2 S-RBD antibody titers between study participants who had and had not received a DT booster were analyzed using the Mann–Whitney test. Differences in anti-SARS-CoV-2 S-RBD titers or SARS-CoV-2-specific T cell-derived IFN-γ concentrations among the groups was assessed using the Kruskal–Wallis test. If the difference was significant (*p*-value less than 0.05), Dunn’s multiple comparisons test was performed to analyze the differences between the two groups. The data were visualized using GraphPad Prism version 10.2.3 (GraphPad Software, Boston, MA, USA).

## 3. Results

### 3.1. Characteristic of Study Participants

One hundred and fifty four children were recruited for this study (Table 1), with a median age of 92 months and BMI of 14.8 kg/m^2^, which is considered underweight [49]. The proportion of boys and girls were 39% and 61%, respectively. Most study participants came from lower-income families. Although a prior diagnosis of COVID-19 excluded eligible subjects from this study, most study participants reported a history of unknown acute respiratory infection in the past 6 months, of whom, 31.2% experience a respiratory infection more than three times. Many of them also had family members who had been diagnosed with COVID-19.

The study participants were subsequently classified based on their COVID-19 and DTP vaccination statuses. The study participants were categorized as “yes” for COVID-19 vaccination if they had received two doses of the CoronaVac vaccine; the participants were categorized as “yes” for DTP vaccination if they had received three doses of the DTP vaccine and one dose of the DT booster vaccine at 5 years old, irrespective of whether they had received an additional dose of the DTP vaccine before 2 years old. The study participants were subsequently grouped into four groups: there were 39 children in group A (“COVID-19 yes/DTP yes”), 38 children in group B (“COVID-19 yes/DTP no”), 38 children in group C (“COVID-19 no/DTP yes”), and 39 children in group D (“COVID-19 no/DTP no”). The frequency of acute respiratory infection as well as incidence of COVID-19 among their family members were evenly distributed among these four groups (Appendix A).

### 3.2. Humoral Immune Response Following DTP and/or COVID-19 Vaccination

DTP primary vaccination (using whole-cell pertussis) along with DT booster vaccination are included in the mandatory national vaccination program in Indonesia. In order to assess whether DTP vaccination could enhance the efficacy of CoronaVac vaccination in generating COVID-19-specific immune responses, the study participants were classified into two groups, i.e., “DTP yes” and “DTP no” (i.e., only received three doses of the DTP vaccine). There were 77 children in each group. As a surrogate marker of DTP vaccination + booster status, the titer of anti-diphtheria toxoid IgG was measured in all study participants (Appendix A). As expected, the study participants in the group “DTP yes” had a significantly higher titer (median = 0.9349 IU/mL) compared to the ones in the group “DTP no” (median = 0.2113 IU/mL). Of note, a further sub-stratification of group “DTP yes” based on whether they received an additional dose of the DTP vaccine before 2 years old (“Full”; *n* = 63; median = 0.9349 IU/mL) or not (“Partial”; *n* = 14; median = 0.6290 IU/mL) did not show any significant difference (*p* = 0.2630; Appendix A). The group “DTP yes” in the subsequent analyses consisted of all study participants receiving the DT booster vaccine, irrespective of whether they had received an additional dose of the DTP vaccine before two years old.

The study participants were subsequently assessed based on their COVID-19 vaccination status; they were classified into four groups based on their DTP and COVID-19 vaccination statuses (Appendix A). Regarding the subjects who had received COVID-19 vaccination (*n* = 77), the interval between the second dose of CoronaVac and the laboratory measurements of their humoral and cellular immune responses varied between 12 and 20 months, with the time intervals of 18 (*n* = 27), 17 (*n* = 15) and 19 (*n* = 13) months being frequently observed (Figure 1).

The overall findings suggested that there was no statistical difference in the titers of anti-SARS-CoV-2 S-RBD antibodies among the four groups (*p* = 0.089). Nonetheless, as expected, group A (“COVID-19 yes/DTP yes”) had the highest median titer of anti-S-RBD antibodies (median = 1196 U/mL), while group D (“COVID-19 no/DTP no”) had the lowest median titer of anti-S-RBD antibodies (median = 527.9 U/mL). Group C (“COVID-19 no/DTP yes”) surprisingly had the second highest median titer of anti-S-RBD antibodies (median = 1163 U/mL), despite the participants not receiving two doses of the CoronaVac vaccine. Interestingly, group A had a higher median titer than group B (“COVID-19 yes/DTP no”; median = 771.2 U/mL), suggesting an enhancing effect of DTP vaccination on the CoronaVac vaccine in generating COVID-19-specific humoral immunity. A follow-up analysis was performed by stratifying the titers of anti-SARS-CoV-2 S-RBD antibodies based on their DTP vaccination status. Interestingly, Table 2 shows that the study participants from the group of “DTP yes” had a higher median titer of anti-SARS-CoV-2 S-RBD antibodies than the ones from the group of “DTP no”.

### 3.3. Cellular Immune Response Following DTP and COVID-19 Vaccinations

Upon the observation that DTP vaccination might be able to modulate COVID-19-specific humoral immunity, it was of interest to assess the COVID-19-specific cellular immunity as well. The interferon gamma-release assay following ex vivo whole blood stimulation with a pool of SARS-CoV-2 spike protein-derived peptides was deployed because this approach is relatively simple but reliable and it allows for an unbiased analysis of the T cell response to SARS-CoV-2-specific antigens using limited amounts of blood [48,50]. As shown in Figure 2, the stratification of the study participants based on their COVID-19 and DTP vaccination statuses provided a clear assessment of their COVID-19-specific cellular immunity. Group A had the highest median concentration of T cell-derived IFN-γ, followed by groups B, C, and D. The IFN-γ concentration in group A (median = 560.9 mIU/mL) was significantly higher than that of group C (median = 230.8 mIU/mL; *p* = 0.0003) or D (median = 187.9 mIU/mL; *p* = 0.0027), suggesting a potential synergism between DTP and CoronaVac vaccinations to generate COVID-19-specific T cell immunity in these pediatric subjects. Despite no significant difference in the IFN-γ concentration between groups A and B (median = 560.9 versus 318.0 mIU/mL; *p* = 0.6634), it was interesting to note that there was an increasing trend in IFN-γ production among the study participants who had received both COVID-19 and DTP vaccinations. This finding suggests that complete DTP vaccination might be able to boost the SARS-CoV-2-specific cellular immunity generated by CoronaVac vaccination.

## 4. Discussion

Here, we reported a potential synergistic impact of DTP vaccination + booster on SARS-CoV-2-specific adaptive immune responses among 154 healthy children aged 6–8 years old who had received two doses of the CoronaVac vaccine in Indonesia. Our findings can be summarized in three points. First, despite DTP vaccination + booster being included in the national pediatric immunization program in Indonesia since 2017 [51], we noticed that only 50% of the study participants had received three doses of the DTP primary vaccine and one dose of the DT booster vaccine at 5 years old, resulting in a substantial difference in the anti-diphtheria toxoid IgG titers between children with a complete DTP vaccination status and the ones with an incomplete status. Several factors contributed to the low coverage of DTP vaccination, including inadequate health services across Indonesia, vaccine hesitancy, and the COVID-19 pandemic [52,53]. This low coverage is alarming because Indonesia had recently encountered several outbreaks of diphtheria and pertussis [52,54,55]. Furthermore, it has been postulated that the DTP vaccine can generate trained immunity in innate immune cells and activate heterologous adaptive immunity against unrelated pathogens [40]. It is unlikely that the heterologous immunity could be sufficiently created if the coverage of DTP vaccination + booster is below the required level for mass vaccination.

Second, we observed that the administration of two doses of the CoronaVac vaccine was effective in generating B and T cell-mediated immune responses against SARS-CoV-2 among the study participants. To date, two pediatric COVID-19 vaccines for 6-year-old children and above are approved in Indonesia: CoronaVac and BNT162b2 [26,56]. However, as the latter vaccine has not yet been included in the national childhood vaccination program in Indonesia thus far, we focused our analysis on healthy children who received the former vaccine. In Indonesia, the CoronaVac vaccine is administered twice (at 3 μg per 0.5 mL) with an interval of 28 days for children aged 6–17 years old without any boosters. We indeed observed that the CoronaVac-vaccinated study participants had noticeably higher levels of anti-SARS-CoV-2 S-RBD antibodies and T cell-derived IFN-γ compared to the ones who did not receive COVID-19 vaccination. These findings are in accordance with previous studies on healthy children and adults, showing that CoronaVac, an inactivated SARS-CoV-2 vaccine, could generate both humoral and cellular immune responses [26,47,57,58,59]. While the induced titers of anti-SARS-CoV-2 S-RBD antibodies were relatively low (presumably due to the lower immunogenicity of inactivated SARS-CoV-2 vaccines [20] and a long duration between the second dose of CoronaVac and the blood testing because a substantial waning of the specific antibodies commonly occurs 3 months post-vaccination), the T cell-specific response upon stimulation with peptides of the SARS-CoV-2 spike protein was well maintained in this study, suggesting that an inactivated SARS-CoV-2 vaccine was better at generating specific cellular immunity that would help to protect against the development of severe COVID-19 compared to inducing specific humoral immunity that could prevent a symptomatic SARS-CoV-2 infection [58,60]. Of note, it was recently published that upon administration of two doses of BNT162b2, healthy children aged 5–12 years old mounted antibody, B cell, and T cell responses, in which they had stronger antibody and T cell responses than adults 6 months after vaccination. Importantly, that study also suggested that the T cell response was the most important predictor of protection against COVID-19 in children [61].

Third, we suggest that complete DTP vaccination + booster (i.e., three doses of the DTP primary vaccine and one dose of the DT booster vaccine at 5 years old, irrespective of whether they received an additional dose of the DTP vaccine before 2 years old) might be able to enhance SARS-CoV-2-specific humoral and cellular immune responses. Despite the low coverage of DTP vaccination among the study cohort, we observed that children with complete DTP vaccination + booster exhibited higher titers of anti-SARS-CoV-2 S-RBD antibodies than the ones with an incomplete DTP vaccination + booster status (*p* = 0.026). Although the subsequent stratification based on the COVID-19 and DTP vaccination statuses did not show any significant differences in the anti-SARS-CoV-2 S-RBD antibody titers among the four groups, we speculate that the insignificant difference was presumably due to the relatively weak ability of CoronaVac vaccination to generate higher titers of anti-SARS-CoV-2 S-RBD antibodies, as well as the substantial waning of specific antibodies several months post-vaccination, as was reported by our group and others in healthy children and adults [26,47,62,63,64,65,66]. Another possibility was that some study participants might have been recently be infected with SARS-CoV-2, and hence anti-SARS-CoV-2 S-RBD antibodies were detected in the unvaccinated groups, contributed to the insignificant difference among the four groups. Although we excluded children with a confirmed diagnosis of COVID-19, we could not exclude a possibility that asymptomatic COVID-19 subjects were recruited into this study [5,67]. A future study should be conducted with a large cohort to validate the current findings. In contrast, we observed a significant difference in the T cell-specific IFN-γ concentrations among the four groups, suggesting that a combination of COVID-19 and DTP vaccinations (i.e., group A) could enhance T cell immunity against SARS-CoV-2. The potential role of heterologous immunity was supported by our observation that the IFN-γ concentrations among groups B (“COVID-19 yes/DTP no”), C (“COVID-19 no/DTP yes”), and D (“COVID-19 no/DTP no”) were not significantly different. This idea aligns with published studies, which reported an association between a recent history of DTP vaccination and protection against COVID-19 [42,68]. However, as the IFN-γ concentrations of groups A and B in this study did not differ significantly, this also suggests that either (i) the heterologous immune response due to DTP and CoronaVac vaccinations might not be strong enough to substantially enhance the T cell immunity against SARS-CoV-2 [40,69,70] and receiving the DT vaccine (instead of DTP) as a booster at 5 years old might not be the most optimum choice to generate a heterologous immune response against SARS-CoV-2 [41,68], or (ii) the longer and varied duration between the DT booster vaccination, CoronaVac vaccination, and the immunological assays might have decreased the impact of heterologous immunity in this study [68]. Although several findings of this study were contradictory, we nevertheless suggest that there might be a potential role of heterologous immunity, which could explain the synergistic effect of DTP and CoronaVac vaccinations on SARS-CoV-2-specific adaptive immune responses. In particular, the ex vivo whole blood stimulation assay with SARS-CoV-2 spike protein-derived peptides provided an opportunity to observe the specific production of T cell-derived IFN-γ among the study participants. Group A had the highest median IFN-γ concentration among all the groups in this study; this suggests a usefulness of the childhood routine vaccination program (i.e., DTP vaccination) in inducing heterologous immunity against SARS-CoV-2 antigens, which might be able to enhance the T cell immune responses generated upon vaccination with an inactivated SARS-CoV-2 vaccine in healthy children (Figure 3).

Heterologous immunity is mediated by memory T cell responses induced by a particular pathogen, which could be also directed against another pathogen. This could result in enhanced cellular and humoral adaptive immunities against a novel pathogen [71,72]. This hypothesis is an attractive concept especially when there is an outbreak of a novel pathogen and there is a lack of effective vaccines to control the outbreak due to ongoing research and development or due to global distribution inequality of vaccines [3,4]. Since routine vaccination (particularly childhood vaccinations) has become an integral part of national public health programs across the globe, including Indonesia, the capability of routine vaccines (e.g., DTP, BCG, or MMR) to generate heterologous immunity against COVID-19 has been investigated. Several in silico studies reported that certain pathogenic bacteria, including those that cause diphtheria, tetanus, and pertussis, could generate cross-reactive immunity to SARS-CoV-2 because these bacteria and SARS-CoV-2 share multiple epitopes, including numerous epitopes recognized by CD8^+^ and CD4^+^ T cells, generating broad protection coverage [41,73]. These findings were substantiated by other studies, which reported that elderly people with registered diphtheria or tetanus vaccinations were less likely to develop severe COVID-19 in the UK and USA [42,68]. The in vitro data also suggested a strong correlation between T cell responses to Tdap and SARS-CoV-2 antigens [68]. However, another study reported a conflicting result, showing that DTP vaccination was not associated with a lower incidence of SARS-CoV-2 infection across various age groups in the USA [69]. This difference could be attributed to (i) T cell immunity, as the primary result of heterologous immune response is protection against the development of severe COVID-19 instead of preventing SARS-CoV-2 infection [58,60]; (ii) the Tdap vaccine is commonly used as a booster every 10 years in developed countries despite the recent finding that acellular pertussis antigens (in contrast to whole-cell pertussis antigens) only share a few epitopes with SARS-CoV-2 antigens [41]; (iii) the time interval between routine vaccination, COVID-19 vaccination (or even SARS-CoV-2 infection), and immunological testing; and (iv) the selected age groups (e.g., older individuals might have lower immunity than younger individuals).

Our study therefore had an advantage by focusing on DTP and CoronaVac vaccinations among healthy children aged 6–8 years old. As DTP vaccination and a DT booster at 5 years old are part of the national childhood vaccination program, we were able to narrow the time interval between the DT booster and CoronaVac vaccination (administered as early as 6 years old) to increase the probability of inducing heterologous immunity. Our study demonstrated additional benefits of the routine childhood vaccination program, i.e., vaccination against certain bacteria might be able to boost immune responses against a novel virus. This emphasizes the rationale for and the importance of continuing vaccinating programs against common pathogens, despite the disruption of routine childhood vaccination program due to the pandemic [74]. Furthermore, our study compared the production of specific antibodies and IFN-γ against SARS-CoV-2 to discern any enhancement of DTP vaccination on CoronaVac vaccination efficacy in generating SARS-CoV-2-specific adaptive immunity. This is important because many published studies only measured humoral immunity (e.g., titers of specific antibodies) against COVID-19, resulting in an incomplete analysis and understanding of adaptive immune responses against SARS-CoV-2.

Our study had several limitations. The nature of the cross-sectional study did not allow us to determine any causality between DTP and CoronaVac vaccinations. We also did not measure any clinical outcomes of the study participants pertaining to SARS-CoV-2 infection and/or COVID-19 severity. Nevertheless, two published studies in Chile reported the effectiveness of CoronaVac vaccination among children aged 3–16 years old in protecting against severe COVID-19 due to either the Delta or Omicron variant [75,76], suggesting that CoronaVac vaccination among healthy children in Indonesia might result in a similar efficacy. In addition, the relatively long duration between the second dose of a COVID-19 vaccine and the immunological testing in this study might obscure the distinction of COVID-19 vaccine-related effects among the tested groups, particularly on the titers of anti-SARS-CoV-2 S-RBD antibodies. Although we only recruited healthy children aged 6–8 years old without any confirmed history of COVID-19, we could not exclude the possibility that the study participants might have contracted COVID-19 without exhibiting any serious symptoms or without being diagnosed. This was possible because SARS-CoV-2 infection among pediatric patients is usually milder than the infection in adults [5,67]. Next, CoronaVac, as an inactivated SARS-CoV-2 vaccine, would generate spike-, membrane-, as well as nucleoprotein-specific T cells (particularly CD4^+^ T cells) [58,59]. Thus, by only using peptide pools from the spike protein, we were not be able to assess the overall T cell immune response against all SARS-CoV-2 antigens. Lastly, our study only measured secreted IFN-γ as a marker of the T cell response in vaccinated children. This mono-functionality overlooks the complete capability of virus-specific T cells because SARS-CoV-2-specific T cells appear to be multi-functional, secreting various cytokines (e.g., IL-2 or TNF-α in addition to IFN-γ) [50,77]. The limited blood volume from the children also did not allow us to investigate antigen-specific T cells using flow cytometry. Nonetheless, a recent study on healthy children aged 5–12 years old vaccinated with the BNT162b2 vaccine reported that the T cell response, as indicated by IFN-γ release, was the most important predictor for protection against symptomatic SARS-CoV-2 infection [61].

## 5. Conclusions

In this study, we analyzed the potential of complete DTP vaccination + booster in generating heterologous immunity that could enhance the SARS-CoV-2-specific adaptive immunity that was induced by an administration of an inactivated SARS-CoV-2 vaccine among healthy children aged 6–8 years old. Upon stimulation with peptides of the SARS-CoV-2 spike protein, we observed higher concentrations of IFN-γ secreted by T cells in study participants with complete DTP and CoronaVac vaccination statuses, which could provide protection against COVID-19 in pediatric populations. Our findings support the usefulness of pediatric COVID-19 vaccination as it generated SARS-CoV-2-specific adaptive immune responses, and could therefore serve as an impetus for policymakers in Indonesia to continuously provide COVID-19 vaccines for children. Our results suggest that DTP vaccination + booster might be useful in enhancing SARS-CoV-2-specific immune responses among children who received an inactivated COVID-19 vaccine. This generation of protection against novel pathogens could become an additional benefit of routine childhood vaccination (e.g., DTP vaccination). However, this finding needs to be confirmed in prospective studies using larger pediatric cohorts. Our study also emphasizes the importance of policymakers maintaining routine childhood immunization despite an epidemic. Further studies will be required to confirm and elucidate the exact mechanism of this heterologous adaptive immunity (including determining the exact cross-reactive epitopes) and to enhance it in order to provide protection against novel pathogens.

## Figures and Tables

**Figure 1 vaccines-12-01032-f001:**
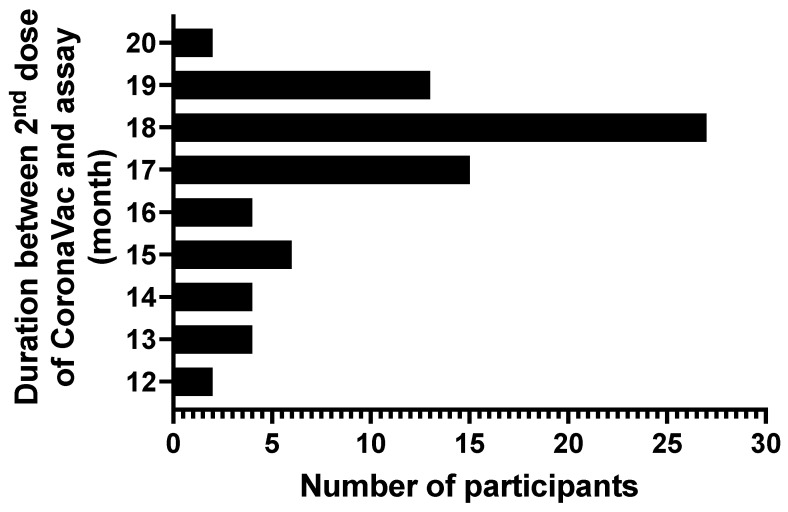
Time interval between the second dose of CoronaVac vaccine and laboratory assays measuring SARS-CoV-2-specific humoral and cellular immune responses. The data were obtained from 77 subjects who received DTP vaccination + booster and CoronaVac vaccination. The vertical axis is the time interval in months, and the horizontal axis is the absolute number of participants.

**Figure 2 vaccines-12-01032-f002:**
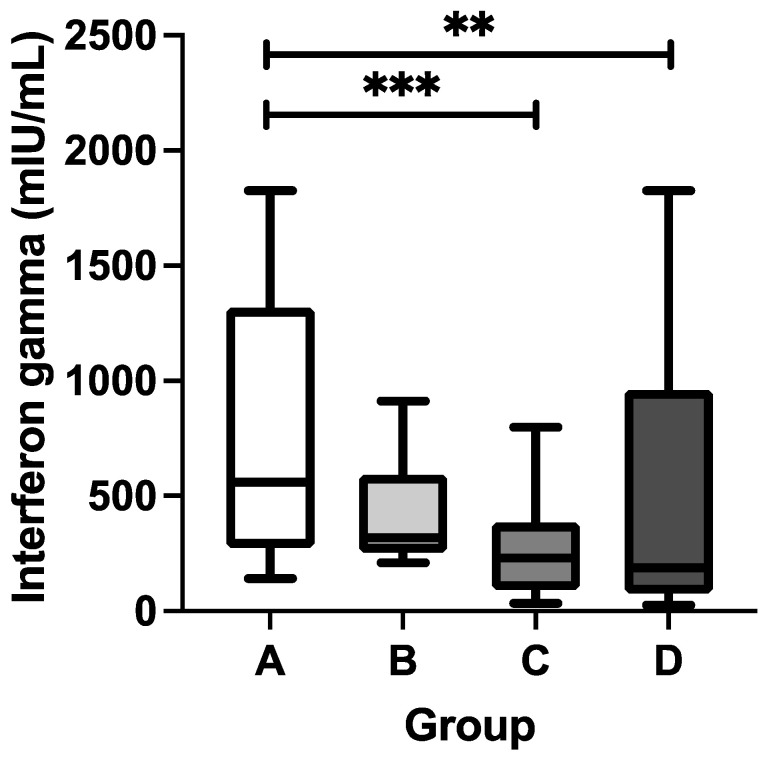
Concentration of SARS-CoV-2-specific T cell-derived interferon gamma in serum of study participants. The study participants were classified into four groups based on their COVID-19 and DTP vaccination statuses: A (“COVID-19 yes/DTP yes”), B (“COVID-19 yes/DTP no”), C (“COVID-19 no/DTP yes”), and D (“COVID-19 no/DTP no”). The solid horizontal line within each box refers to the median value. The whiskers refer to the 10 and 90 percentile values, respectively. The Kruskal–Wallis test was performed to determine if there was a statistical difference among the four groups. If it was significant (*p* < 0.05), Dunn’s multiple comparisons test was subsequently performed. The *** and ** indicate *p* < 0.001 and *p* < 0.01, respectively.

**Figure 3 vaccines-12-01032-f003:**
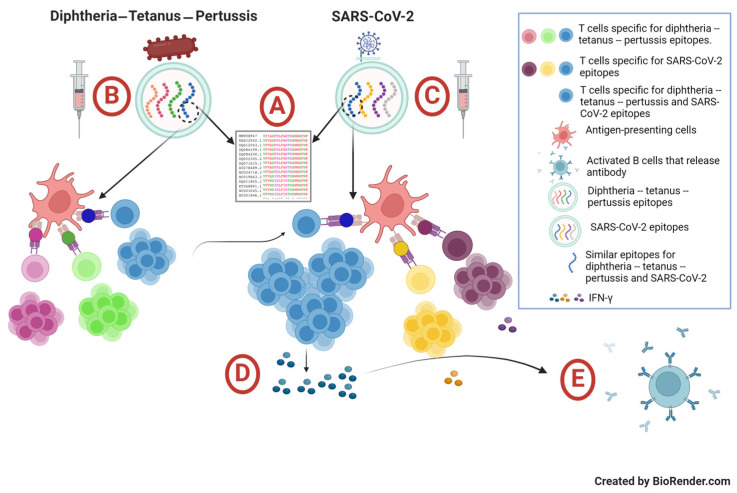
Proposed mechanism of heterologous immunity induced by diphtheria–tetanus–pertussis vaccination in enhancing immune responses generated by inactivated SARS-CoV-2 vaccines in healthy children. (**A**) There are similar epitopes shared by the diphtheria–tetanus–pertussis vaccine and inactivated SARS-CoV-2 vaccine targets, particularly the SARS-CoV-2 spike protein. (**B**) Upon administration of the diphtheria–tetanus–pertussis vaccine or diphtheria–tetanus booster, certain peptide antigens will form peptide–MHC complexes to stimulate diphtheria-/tetanus-/pertussis-specific T cells. (**C**) Subsequently, upon administration of an inactivated SARS-CoV-2 vaccine, certain diphtheria-/tetanus-/pertussis-specific T cells will respond to certain antigens of the SARS-CoV-2 spike protein. This will generate a secondary immune response, resulting in heterologous immunity. (**D**) The heterologous immunity would also stimulate SARS-CoV-2 spike protein-specific T cells to produce various cytokines, including interferon gamma, at higher concentrations. (**E**) The activated CD4^+^ T cells would stimulate specific B cells to mature and release immunoglobulins. This figure was created with BioRender.com.

**Table 1 vaccines-12-01032-t001:** Characteristics of study participants (*n* = 154).

Variable	Value
Age, months [median (minimum–maximum)]	92 (81–103)
BMI, kg/m^2^ [median (minimum–maximum)]	14.8 (12.3–19.7)
Sex [*n* (%)]	
Male	60 (39)
Female	94 (61)
Parental occupation [*n* (%)]	
Working	71 (46.1)
Not working	83 (53.9)
Parental income [*n* (%)]	
Equal or above the minimum wage	24 (15.6)
Below the minimum wage	130 (84.4)
History of acute respiratory infection in last 6 months [*n* (%)]	
<3 times	106 (68.8)
≥3 times	48 (31.2)
History of COVID-19 disease in other family members [*n* (%)]	
No	73 (47.4)
Yes	81 (52.6)
Classification based on vaccination statuses [*n* (%)]	
Group A (COVID-19 yes/DTP yes)	39 (25.3)
Group B (COVID-19 yes/DTP no)	38 (24.7)
Group C (COVID-19 no/DTP yes)	38 (24.7)
Group D (COVID-19 no/DTP no)	39 (25.3)

The minimum wage in Jakarta in 2024 was IDR 5,067,381 (approximately USD 325). COVID-19 vaccination status was recorded as a yes if the subject received 2 doses of CoronaVac. DTP vaccination status was recorded as a yes if the subject received 3 doses of the DTP vaccine and a booster DT vaccine at five years old (with or without receiving an additional DTP vaccine before two years old). BMI, body mass index; COVID-19, coronavirus disease 2019; DTP, diphtheria–tetanus–pertussis; DT, diphtheria–tetanus.

**Table 2 vaccines-12-01032-t002:** Titers of anti-SARS-CoV-2 S-RBD antibodies based on DTP vaccination status.

DTP Vaccination Status	*n*	Anti-SARS-CoV-2 S-RBD (U/mL) Median (Min–Max)	*p*-Value
Yes	77	1182 (0.3–22,269)	0.026
No	77	612.5 (0.3–14,589)	

Mann–Whitney test was performed with *p*-values < 0.05 considered statistically significant.

## Data Availability

Data are available upon request to the corresponding author.

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
