# Peer review of "The Improvement of Adaptive Immune Responses towards COVID-19 Following Diphtheria–Tetanus–Pertussis and SARS-CoV-2 Vaccinations in Indonesian Children: Exploring the Roles of Heterologous Immunity"

_vaccines, 2024, doi:10.3390/vaccines12091032_

Round 1

Reviewer 1 Report

Comments and Suggestions for Authors

Dear Editor,

Dear Author,

I did like to read the present paper submitted to vaccines.

The Improvement of Adaptive Immune Responses towards COVID-19 Following Diphtheria-Tetanus-Pertussis and SARS-CoV-2 Vaccinations in Indonesian Children: Exploring Roles of Heterologous Immunity

It is an interesting study, but the results are somehow contradictory. Anti-SARS-CoV-2 antibodies in vaccinated were similar as they were in the no vaccinated children. How could this be?

Results such as complete DPT is better than incomplete vaccination is propaedeutic.

1. I would encourage the authors to report on the safety administrating the regime of vaccination in children. This would be beneficial.

 2. I would structure the manuscript differently, as it is currently. I would focus on the main findings.

3. Even the authors try to explain the results; I would also discuss them as a potential artefact, as they are in part inconclusive.

4. I recommend to temper phrases which outline the DPT would enhance the immunization against COVID.

Author Response

Comments 1: It is an interesting study, but the results are somehow contradictory. Anti-SARS-CoV-2 antibodies in vaccinated were similar as they were in the no vaccinated children. How could this be? Results such as complete DPT is better than incomplete vaccination is propaedeutic.

Response 1: We thank Reviewer #1 for the constructive comments. It is an important remark, which we have attempted to discuss. In the revised manuscript, we proposed several possibilities that could partly explain the inconclusive findings of COVID-19-specific humoral immunity. First, we could not exclude a possibility that some study participants might had already been infected with SARS-CoV-2 but showing no or mild symptoms, hence those children had never been clinically diagnosed with COVID-19 (i.e., an exclusion criterion). It would certainly generate anti-SARS-CoV-2 antibodies in those subjects, despite they have not been vaccinated. This has been included in the revised manuscript (line 432-438 & 534-538). Second, we include new data of duration between the second dose of CoronaVac vaccination and timing of antibody’s measurement, reporting that sera titers of anti-SARS-CoV-2 S-RBD antibodies in majority of CoronaVac-vaccinated children in our study (n=55 out of 77) were measured around 17-19 months post-vaccination. As several research groups had reported that titer of anti-SARS-CoV-2 S-RBD antibodies steadily declined over 6-9 months (e.g., doi:10.1126/science.abf4063; doi:10.3390/v15040917; doi:10.1016/j.cell.2022.05.022), our vaccinated study participants arguably would no longer display high titers of anti-SARS-CoV-2 S-RBD antibodies. This has been included in the revised manuscript (line 308-312, 335-340 and 531-534). Third, inactivated COVID-19 vaccines, such as CoronaVac, is not a strong inducer or COVID-19-specific humoral immunity, as compared to mRNA-based vaccines. This has been reported by many studies, including ours (e.g., doi:10.1007/s12519-022-00680-9; doi:https://doi.org/10.1016/j.ebiom.2022.103972; doi:10.7774/cevr.2022.11.1.116; doi: 10.7774/cevr.2022.11.2.209), indicating that the peaks of anti-SARS-CoV-2 antibodies upon inactivated COVID-19 vaccination were lower than the ones upon mRNA-based COVID-19 vaccination (line 406-407) Together with a shorter durability of COVID-19-specific humoral immunity upon CoronaVac vaccination, this might partly explain why we observed insignificant difference of titers of anti-SARS-CoV-2 S-RBD antibodies among groups in our study. Four, it has been suggested that inactivated COVID-19 vaccines might be better in generating specific cellular immunity that would help to protect against development of severe COVID-19, rather than inducing specific humoral immunity that would help to prevent a symptomatic SARS-CoV-2 infection (line 409-414). We therefore would like to suggest that for studies that using inactivated COVID-19 vaccines, e.g., CoronaVac vaccine, it would be better to also assess the T-cell immunity as this response is better preserved than the humoral immunity. 

Comments 2: 

I would encourage the authors to report on the safety administrating the regime of vaccination in children. This would be beneficial.

Response 2: 

We agree with this suggestion. In the revised manuscript, we now state the safety of administrating both CoronaVac and DTP/DT vaccinations in children, as can be found in INTRODUCTION line 105-136 and MATERIALS AND METHODS line 192-214.

Comments 3: 

I would structure the manuscript differently, as it is currently. I would focus on the main findings.

Response 3: 

We restructure our manuscript by focussing to report main findings (i.e., COVID-19 adaptive immune responses) in the main text, as can be seen in description of Figure 1 & 2 as well as Table 2 in RESULTS (line 289-373). We also relocate the initial Table 2 (“Titers of anti-SARS-CoV-2 S-RBD antibodies among the study participants”) to Supplementary Table 2. We now briefly discuss results on DTP vaccination, as we combine the description of anti-diphtheria immunoglobulin G with the one of anti-SARS-CoV-2 S-RBD antibodies. We also move results of DTP vaccination to Supplementary Figure 1 & 2.

Comments 4: Even the authors try to explain the results; I would also discuss them as a potential artefact, as they are in part inconclusive.

Response 4: 

In the initial manuscript, we described several limitations of our findings, which might contribute to some inconclusive findings in this study. As mentioned in point #1, we re-emphasize those limitations in the revised manuscript, e.g., duration between the second dose of CoronaVac and timing of laboratory assays as well as a plausibility of recruiting asymptomatic SARS-CoV-2-infected children into our cohort (line 534-538). We also state that “Another possibility was that some study participants might recently be infected with SARS-COV-2, hence anti-SARS-CoV-2 S-RBD antibodies were detected in the unvaccinated groups, contributed to the insignificant difference among four groups. Although we had excluded children with confirmed diagnosis of COVID-19, we could not exclude a possibility that asymptomatic COVID-19 subjects were recruited into this study [5,67]. The future study should be conducted as a large prospective cohort study to validate the current findings” in DISCUSSION line 432-438. We further discuss it in line 453-460 by stating “Although several findings of this study were inconclusive, we nevertheless suggest that there might be a potential role of heterologous immunity, which could explain the synergistic effect upon DTP and CoronaVac vaccinations on SARS-CoV-2-specific adaptive immune responses. In particular, the ex vivo whole blood stimulation assay with SARS-CoV-2 spike protein-derived peptides provided an opportunity to observe specific production of T cell-derived IFN-y among study participants”.

Comments 5:

I recommend to temper phrases which outline the DPT would enhance the immunization against COVID.

Response 5: 

In the revised manuscript, we moderate phrases that outlined DTP vaccination would enhance the immunization against COVID-19, e.g., in line 41, 346, 357-358, 363, 376, 410, 421-422, 454-456 and 462. We also revise our CONCLUSIONS to mention “Our results suggest that DTP vaccination-booster might be useful to enhance SARS-CoV-2-specific immune responses among children who received inactivated COVID-19 vaccine. This could become an additional benefit of routine childhood vaccination (e.g., DTP) to generate protection against novel pathogens. However, this finding needs to be reconfirmed in prospective studies using larger pediatric cohorts. Our study also emphasizes the importance for policymakers to maintain routine childhood immunization despite of new epidemic” (line 562-568).

Reviewer 2 Report

Comments and Suggestions for Authors

The manuscript seeks to explore the concept of heterologous immunity. In this phenomenon, the immune response to a pathogen is influenced by a previous exposure to a different pathogen in children receiving both DTP and inactivated SARS-CoV-2 vaccines. The study suggests that this combined vaccination may lead to a greater B (IgG concentration) and T (IFN-Y) immunity. However, the work's rationale is based on unproven facts from the literature, does not significantly advance knowledge on the subject, and presents several postulation contradictions. Therefore, we do not recommend its publication.

Major points

1) MM-Inform the type of DTP vaccine used and the manufacturer, as this information is crucial for the credibility and reproducibility of the study. Different DTP vaccines may have varying effects on the immune system, and knowing the specific type and manufacturer used in the study can help other researchers replicate the results or conduct further investigations.

2) Line 300-"We observed that administering two doses of CoronaVac vaccine effectively generated B and T cell-mediated immune responses towards SARS-CoV-2 among the study participants. - This is already a consolidated fact in the literature and does not bring anything new.

3) Line 313-321-While the induced titers of anti-SARS-CoV-2 S-RBD antibodies were relatively low, the T cell-specific response upon stimulation with peptides of SARS-CoV-2 spike protein was well maintained in this study. This supports the idea that the inactivated SARS-CoV-2 vaccine may be better at generating specific cellular immunity to protect against severe COVID-19, rather than inducing specific humoral immunity to prevent symptomatic SARS-CoV-2 infection. However, this sentence is quite long and difficult to understand, and it's unclear what the authors are trying to convey.

4) Line 360-361- Heterologous immunity is only T cell-response?

5) Lines 381-383- I need clarification on how simply measuring the level of Immunoglobulin and If-Y in children vaccinated with DTP and COVID can indicate cause and effect in the group of double-vaccinated children. In the same sense, how can we explain an increase in the concentration of immunoglobulins induced by a few epitopes capable of being measured spectrophotometrically?

6) Further studies will be required to elucidate the exact mechanism of heterologous adaptive immunity and enhance it as a priming to protect against novel pathogens. At what point did the work contribute to elucidating the fact that much more studies are necessary?

7) There is no information on the post-vaccination collection time of sera between the different vaccinated groups (both those vaccinated with DTP and COVID-19). Information is important between the two groups and can help clarify the small difference in the concentration/measurement of immunoglobulins.

8) The work cited in the study (doi:10.3389/fimmu.2020.586984) describes only 2 Spike protein peptide sequences [YNENG (S1 region) and LITGRQS (S2R1 region] that could putatively be considered cross-reactive B epitopes using bioinformatics approaches but have not been described (by others) as valid epitopes using experimental results and are not located in the RBD motif.

9) How to explain the protective importance of these two cross-reactive epitopes in a universe of 6 IgG epitopes present in the RBD motif (cellular receptor binding site) and 42 throughout the entire protein length.

10) In all studies, there isn’t information on which bacterial protein cross-reacts to the SARS-CoV-2 spike protein. Please check the paper (Toxins 2023, 15, 239. https://doi.org/10.3390/toxins15040239).

11) How can we explain the increased protective effect of children double vaccinated with DTP x SARS-CoV-2 during a post-pandemic pertussis resurgence in many countries?

Comments on the Quality of English Language

In many parts, English is difficult to understand. Grammatical and sentence construction errors should be improved.

Author Response

Comments 1: 

However, the work's rationale is based on unproven facts from the literature, does not significantly advance knowledge on the subject, and presents several postulation contradictions. Therefore, we do not recommend its publication.

Response 1: 

The concept of heterologous immunity is interesting and yet, it is actually an old concept. Heterologous immunity is a term describing the phenomenon how an infection/vaccination with one specific pathogen influences innate and adaptive immune responses to an unrelated pathogen. Several examples on this phenomenon are: (i) the first vaccine against smallpox was resulted from cowpox immunization; (ii) an observation that inoculation with malaria could be used to treat syphilis; (iii) HCV infection can have significant imprints on the immune response of other pathogens, as well as (iv) activated herpesvirus-specific CD8+ T cells could contribute to the heterologous anti-viral T-cell response. (PMID:20316823; doi:10.1016/j.coviro.2016.01.005; doi:10.1371/journal.ppat.1001051). Pertaining to a concept of heterologous immunity between anti-bacterial vaccinations and SARS-CoV-2, several studies have also provided supporting findings (doi:10.3389/fimmu.2020.586984; doi:10.1016/j.medj.2021.08.004; doi:10.7189/jogh.13.06004; doi:10.3389/fimmu.2021.749264). Taken together, this suggests that our work’s rationale is not based on unproven facts from the literature. Saying this, more studies are required to confirm the existence of heterologous immunity and to subsequently explain its mechanism (doi:10.3389/fimmu.2020.0140). In many cases, science is advanced by an accumulation of small improvements from time to time (and not due to a revolution of idea), which could also increase the attractiveness of the research topic, hence more research groups will study it. We therefore suggest that our study could help to advance knowledge on this subject. We also moderate our claims in the revised manuscript to indicate that our study did not establish a causal relationship between DTP/DT and COVID-19 vaccinations but suggested a possible relationship between both vaccinations to enhance immunity against SARS-CoV-2 infection.

Comments 2:

MM-Inform the type of DTP vaccine used and the manufacturer, as this information is crucial for the credibility and reproducibility of the study. Different DTP vaccines may have varying effects on the immune system, and knowing the specific type and manufacturer used in the study can help other researchers replicate the results or conduct further investigations.

Response 2: 

We thank for this suggestion and in the revised manuscript, we have added this information in line 194-196.

Comments 3: 

Line 300-"We observed that administering two doses of CoronaVac vaccine effectively generated B and T cell-mediated immune responses towards SARS-CoV-2 among the study participants. - This is already a consolidated fact in the literature and does not bring anything new.

Response 3: 

The usage of CoronaVac vaccination for pediatric population in Indonesia has just been started from December 2021. Later in October 2022, its availability became very limited (line 138-147). However, there is no academic publication that investigate the safety and immunogenicity of CoronaVac in Indonesian children aged 6-8 years old to our knowledge. Amidst parental hesitancy as well as the government’s fluctuating commitment to vaccinate healthy children in Indonesia, published academic data are crucially required to discuss the safety and usefulness of using inactivated COVID vaccine among Indonesian children. We therefore think that our manuscript can partly help to provide the required information.

Comments 4: 

Line 313-321-While the induced titers of anti-SARS-CoV-2 S-RBD antibodies were relatively low, the T cell-specific response upon stimulation with peptides of SARS-CoV-2 spike protein was well maintained in this study. This supports the idea that the inactivated SARS-CoV-2 vaccine may be better at generating specific cellular immunity to protect against severe COVID-19, rather than inducing specific humoral immunity to prevent symptomatic SARS-CoV-2 infection. However, this sentence is quite long and difficult to understand, and it's unclear what the authors are trying to convey.

Response 4: 

We now elaborate these sentences in the revised manuscript (line 405-414) to better convey our message, i.e., the inactivated COVID-19 vaccine was not a strong inducer of humor immunity but was quite a robust inducer of cellular immunity. Although the activated cellular immunity could not prevent SARS-CoV-2 infection, the specific T-cell response would protect vaccinated subjects against severe disease of COVID-19. Our finding was also in accordance with other published studies, e.g., doi:10.3389/fimmu.2023.1139620 and 10.1016/j.immuni.2022.08.008.

Comments 5: 

Line 360-361- Heterologous immunity is only T cell-response?

Response 5: 

No, heterologous immunity does not only exist as T-cell response. It does primarily activate T-cell response due to similarities between recognized peptide antigens (i.e., cross-reactive epitopes), in which it will also support B-cell activation. Of note, CD4+ T cells support activated B cells to go through phases of isotype switching and affinity maturation of their B-cell receptors, result in production of specific immunoglobulins with higher affinity (https://www.ncbi.nlm.nih.gov/books/NBK27142/). Thus, heterologous immunity could exist as cellular and humoral immune responses. We revise the sentence in line 480-482 to clarify this.

Comments 6: 

Lines 381-383- I need clarification on how simply measuring the level of Immunoglobulin and If-Y in children vaccinated with DTP and COVID can indicate cause and effect in the group of double-vaccinated children. In the same sense, how can we explain an increase in the concentration of immunoglobulins induced by a few epitopes capable of being measured spectrophotometrically?

Response 6: 

As mentioned in point #1, we do not declare that there was a causality relationship between DTP and COVID-19 vaccinations in our report because of several limitations found in our study (line 524-550), particularly the cross-sectional setting of our study did not allow us to further investigate it. In addition, as mentioned in point #5, the concept of heterologous immunity primarily involves T-cell activation through a recognition of cross-reactive epitopes, which subsequently could activate both cellular and humoral responses (doi:10.3389/fimmu.2020.0140). We suggest that this results in, at least, heightened production of antibodies. 

Comments 7: 

Further studies will be required to elucidate the exact mechanism of heterologous adaptive immunity and enhance it as a priming to protect against novel pathogens. At what point did the work contribute to elucidating the fact that much more studies are necessary?

Response 7: 

In the revised manuscript, we clarify this sentence by stating that “Our results suggest that DTP vaccination-booster might be useful to enhance SARS-CoV-2-specific immune responses among children who received inactivated COVID-19 vaccine. This could become an additional benefit of routine childhood vaccination (e.g., DTP) to generate protection against novel pathogens. However, this finding needs to be reconfirmed in prospective studies using larger pediatric cohorts. Our study also emphasizes the importance for policymakers to maintain routine childhood immunization despite of new epidemic. Further studies will be subsequently required to confirm and elucidate the exact mechanism of heterologous adaptive immunity (including determining the exact cross-reactive epitopes) and to enhance it as a priming to protect against novel pathogens” in line 562-571.    

Comments 8:

There is no information on the post-vaccination collection time of sera between the different vaccinated groups (both those vaccinated with DTP and COVID-19). Information is important between the two groups and can help clarify the small difference in the concentration/measurement of immunoglobulins.

Response 8: 

We agree with this remark. In the revised manuscript, we state in line 212-214 that “Of note, timing of administration for DTP vaccine, DT booster vaccine and CoronaVac vaccine were obtained from the official vaccination records“. We also provide a new Figure 1 to describe the exact timing between second dose of CoronaVac vaccination and blood testing for participants who were vaccinated with both DTP and CoronaVac in group A & B (total n = 77; line 335-340), in which its result is described in line 308-312.

Comments 9: 

The work cited in the study (doi:10.3389/fimmu.2020.586984) describes only 2 Spike protein peptide sequences [YNENG (S1 region) and LITGRQS (S2R1 region] that could putatively be considered cross-reactive B epitopes using bioinformatics approaches but have not been described (by others) as valid epitopes using experimental results and are not located in the RBD motif. How to explain the protective importance of these two cross-reactive epitopes in a universe of 6 IgG epitopes present in the RBD motif (cellular receptor binding site) and 42 throughout the entire protein length. In all studies, there isn’t information on which bacterial protein cross-reacts to the SARS-CoV-2 spike protein. Please check the paper (Toxins 2023, 15, 239. https://doi.org/10.3390/toxins15040239).

Response 9: 

This is an interesting point. Reviewer #2 pointed out that in a reference doi:10.3389/fimmu.2020.586984, the author had mentioned only 2 Spike protein peptide sequences (LLRYNENG and LITGHLQS) when discussing potential cross-reactive immunity to SARS-CoV-2 from prevalent viruses and viruses targeted by vaccinations. This is correct because the author had suggested a less likelihood that current pediatric vaccinations for viral diseases could protect against SARS-CoV-2 infection. Nonetheless, the author had reported more potential cross-reactive epitopes (including RLFRKSNL and SFELLHAPAT from whole Pertussis antigen as well as FEYVSQPF from Diphtheria antigen) were detected between current pediatric vaccinations for bacterial diseases and SARS-CoV-2. The author had finally suggested that DTP vaccine could become significant sources of potential cross-reactive immunity to SARS-CoV-2. In an unrelated study (doi:10.1016/j.medj.2021.08.004), Mysore et al had reported direct molecular evidence of high frequency of overlapping T-cell receptors among T-cell clones that respond to SARS-CoV-2 proteins and Tdap antigens (i.e., tetanus and lower concentrations of diphtheria and acellular pertussis as a vaccine used in adults), suggesting that heterologous immunity between anti-bacterial vaccination and viral infection is prevalent in humans.

Pertaining to the exact cross-reactive epitopes, Mysore et al (doi:10.1016/j.medj.2021.08.004) had reported using 5 mg/ml of heat-inactivated pertussis toxin, diphtheria toxin or tetanus toxoid to activate T cells in their study, which were still crude mixtures of proteins (doi:10.1016/j.vaccine.2019.04.059; doi:10.1186/s13104-13019-14373-13102; doi:10.3390/proteomes7020015). We therefore agree with Reviewer #2 to suggest that future studies are also required to determine the exact cross-reactive epitopes between DTP and SARS-CoV-2 antigens (line 570).

Comments 10: How can we explain the increased protective effect of children double vaccinated with DTP x SARS-CoV-2 during a post-pandemic pertussis resurgence in many countries?

Response 10: 

We already highlighted a finding that the DTP/DT vaccination coverage in our cohort was worryingly low, in which only 50% of the study participants had received 3 doses of DTP primary vaccine with 1 dose of DT booster vaccine at 5 years old (line 379-384). Unfortunately, this finding was in line with the actual situation worldwide (https://www.unicef.org/indonesia/press-releases/indonesia-targets-low-vaccination-areas-tackle-decline-childhood-immunization; https://www.who.int/news-room/fact-sheets/detail/immunization-coverage). Thus, the potential protective effect could not be realized, arguably, due to the low coverage of childhood routine vaccination programs and instead resulting in pertussis resurgence in many countries.

Reviewer 3 Report

Comments and Suggestions for Authors

Dear authors,

I have now completed the review of the manuscript titled "The Improvement of Adaptive Immune Responses towards COVID-19 Following Diphtheria-Tetanus-Pertussis and SARS-CoV-2 Vaccinations in Indonesian Children: Exploring Roles of Heterologous Immunity."

In the present study, the authors provided interesting preliminary evidence for potential heterologous immunity.

The manuscript is interesting and, in general, fairly well-written.

I have some suggestions to further improve the quality of the manuscript.

I would like to suggest that the authors address these limitations in the article, either by discussing them in the limitations section or, where feasible, by making the appropriate revisions:

1. The introduction is relatively short, and does not introduce recent comprehensive systematic reviews on immunogenicity of COVID-19 vaccines in patients with diverse health conditions.

2. The long and variable time interval between vaccinations and immune testing makes it difficult to isolate vaccine effects from other factors that may influence immunity over time. Also, I think some key details are missing, such as the exact timing between vaccinations and testing for each participant.

3. Only measuring IFN-γ as a marker of T cell responses provides a limited view of cellular immunity. I suggest authors to measure other cytokines or using flow cytometry would give a more complete picture.

4. How did study rule out the possibility that some participants had asymptomatic COVID-19 infections that boosted their immunity?

Thank you for your valuable contributions to our field of research. I look forward to receiving the revised manuscript.

Author Response

Comments 1: I would like to suggest that the authors address these limitations in the article, either by discussing them in the limitations section or, where feasible, by making the appropriate revisions: The introduction is relatively short and does not introduce recent comprehensive systematic reviews on immunogenicity of COVID-19 vaccines in patients with diverse health conditions.

Response 1: 

In the revised manuscript, we expand the INTRODUCTION by including sections on immunogenicity, vaccine effectiveness, as well as safety of various COVID-19 vaccinations (particularly on BNT162b2 and CoronaVac vaccines) in healthy children and pediatric patients with diverse health conditions (line 81-136).

Comments 2: 

The long and variable time interval between vaccinations and immune testing makes it difficult to isolate vaccine effects from other factors that may influence immunity over time. Also, I think some key details are missing, such as the exact timing between vaccinations and testing for each participant.

Response 2: 

We agree with Reviewer #3 as in the initial manuscript, we already described that the relatively long interval between the second dose of CoronaVac vaccination and timing of laboratory assays was one of the weaknesses of this study. This is also due to the nature of cross-sectional study, which did not allow us to capture the kinetic of COVID-19-specific adaptive immune responses over time. We therefore suggest in the revised manuscript that future studies in a setting of prospective large cohort, will be required to confirm our current speculation (line 432-438 & 568-571). We nevertheless would like to point out that this is the real-life situation affecting many low-to-middle-income nations, in which a scarcity of COVID-19 vaccines for pediatric population (in many cases due to wavering commitment of policymakers to provide free COVID-19 vaccines for children) and parental hesitancy did not allow us to conduct an ideal kind of clinical trial, i.e., a prospective cohort study, which we can vaccinate our pediatric subjects with DTP/DT and CoronaVac vaccines and monitor them in a timely manner. Thus, a learning from this real-life situation might be useful as it could contribute to the implication for further vaccination.

Pertaining to the exact timing of vaccination, we now add a sentence in line 212-214 to state that “Of note, timing of administration for DTP vaccine, DT booster vaccine and CoronaVac vaccine were obtained from the official vaccination records“. In the revised manuscript, we also provide a new Figure 1 to describe the exact timing between second dose of CoronaVac vaccination and blood testing for participants in group A & B (total n = 77; line 335-340), in which its result is described in line 308-312.

Comments 3: 

Only measuring IFN-γ as a marker of T cell responses provides a limited view of cellular immunity. I suggest authors to measure other cytokines or using flow cytometry would give a more complete picture.

Response 3: 

We acknowledge that IFN-y measurement would not be able to provide a full potential of activated COVID-19-specific T cells (line 542-550). However, as our study participants were healthy children aged 6-8 years old, we were only permitted to obtain limited amount of blood (i.e., maximum 4 mL per subject). We therefore chose to perform the interferon-gamma release assay upon specific stimulation using SARS-CoV-2 spike protein-derived peptides because this assay could use a very limited amount of blood to provide a specific analysis on COVID-19-specific T-cell functionality (line 457-459). In addition, parts of the collected blood samples were used to measure anti-diphtheria and SARS-CoV-2 S-RBD antibodies, thus it did not allow us to measure other cytokines or to even perform flow cytometry. Nonetheless, several studies have mentioned that IFN-y can be chosen as a surrogate marker for functional cross-reactive T cells (doi:10.1016/j.heliyon.2023.e17186; doi:10.1172/JCI152379; doi:10.1038/s41587-022-01347-6; doi:10.1016/j.medj.2021.08.004) and even as an important predictor for protection against symptomatic SARS-CoV-2 infection (doi:10.1038/s41591-024-02962-3). Taken together, this supports the usefulness in measuring IFN-y in population-based studies, such as ours.

Comments 4: 

How did study rule out the possibility that some participants had asymptomatic COVID-19 infections that boosted their immunity?

Response 4: 

Despite our effort to only include healthy subjects (by only recruiting children aged 6-8 years old with complete DTP primary vaccination but without confirmed medical history of COVID-19; line 179-180), we could not exclude a possibility that some participants had asymptomatic SARS-CoV-2 infections, as stated in line 432-438 & 534-541. The nature of cross-sectional study also did not allow us to measure anti-SARS-CoV-2 antibodies to exclude those children prior the study. We therefore propose that a future study using a setting of large prospective cohort study should be conducted to validate the current findings (line 440-441 and 566-572).

Round 2

Reviewer 1 Report

Comments and Suggestions for Authors

The authors incorporated the suggestions seriosly. the amended the manuscript accordantly.

I have one suggestion left:

I would replace the word inconclusive in the manuscrip by contradictory. It sounds better and more appropriate....

Author Response

Comment 1: 

The authors incorporated the suggestions seriosly. the amended the manuscript accordantly. I have one suggestion left: I would replace the word inconclusive in the manuscrip by contradictory. It sounds better and more appropriate...

Response 1: 

We thank reviewer #1 for the suggestion. In the revised manuscript we now change the word "inconclusive" by 'contradictory" as can be found in DISCUSSION line 454. We agree that this sounds more appropriate.

Reviewer 2 Report

Comments and Suggestions for Authors

The authors present strong arguments that support the issues raised. A good review of the text has been done and therefore we now suggest its publication.

Comments on the Quality of English Language

Only a few minor corrections are necessary and can be made during editing.

Author Response

Comments 1: The authors present strong arguments that support the issues raised. A good review of the text has been done and therefore we now suggest its publication. Only a few minor corrections are necessary and can be made during editing.

Response 1: We thank Reviewer#2 for the positive comments.

Reviewer 3 Report

Comments and Suggestions for Authors

All comments were addressed.

Author Response

Comment 1: All comments were addressed.

Response 1: We thank Reviewer #3 for the constructive remarks.